# Topological characteristics of international business cycle synchronization: A network analysis of the BRI economies

Zhiping Qiu[1,2], Sichao Mai[3]*

1 Institute of Finance and Economics, Shanghai University of Finance and Economics, Shanghai, Shanghai, P.R. China, 2 School of Urban and Regional Sciences, Shanghai University of Finance and Economics, Shanghai, Shanghai, P.R. China, 3 School of Economics and Management, Nanchang Hangkong University, Nanchang, Jiangxi, P.R. China

* jxncmsc@163.com

**Data Availability Statement:** All relevant data are within the article and its Supporting information files.

**Funding:** The author(s) received no specific funding for this work.

## Abstract

Based on the GDP constant 2010 US$ from the World Bank, this paper uses the instantaneous quasi-correlation coefficient to measure the business cycle synchronization linkages among 53 Belt and Road Initiative (BRI) economies from 2000 to 2019, and empirically studies the topological characteristics of the Business Cycle Synchronization Network (BCSN) with the help of complex network analysis method. The main conclusions are as follows: First, the BCSN density and efficiency of BRI economies are still low, and it presents a topological feature of "small world". Second, the individual characteristics of the economies in the network are obviously different. Among them, China's relative influence is significantly increased, but its betweenness centrality level is still low. Third, since the inception of BRI, the topological characteristics of BCSN of BRI economies have undergone great changes, and their topological evolution has gradually reflected the characteristic of self-stability.

## 1. Introduction

The BRI was initiated by the Chinese president Xi Jinping in 2013, and it was completely introduced during the visit of Kazakhstan and Indonesia. In the background of the history of China's ancient overland and maritime Silk Road, the BRI consists of the Silk Road Economic Belt and the 21st Century Maritime Silk Road, geographically crossing Asia, Europe and Africa. The BRI comprises of over 50 different economies, which cover 80% of globe population and its the estimated cost is over $21.1 trillion US dollars [1]. Since the BRI was proposed in 2013, major achievements have been made in infrastructure construction, international trade, cross-border investment and sustainable growth. The World Bank report (https://www.worldbank.org/en/topic/regional-integration/publication/belt-and-road-economics-opportunities-and-risks-of-transport-corridors) noted that BRI is expected to increase real income by 1.2–3.4% in BRI economies and 0.7%-2.9% globally, with investments that could lift more than 7.6 million people out of extreme poverty. In particular, BRI transport projects have significantly reduced trade costs, which are expected to increase trade in BRI economies by 2.8% to 9.7%, increase in world

**Competing interests:** The authors have declared that no competing interests exist.

trade by 1.7% to 6.2%, and total foreign direct investment in BRI economies by 4.97% [2]. The BRI has strengthened cooperation in trade, investment, infrastructure construction, institutional and cultural exchange among Asia, Europe and Africa, and formed a new economic network [3]. It aims to build a community of shared future for mankind, and it is an important platform for international multilateral cooperation and regional integration cooperation. Ultimately, it will make the world economy and society more open, inclusive, balanced and benefit-sharing. However, there are still significant political, cultural, social, economic and institutional differences among the BRI economies, especially in terms of economic growth and their policies. Such differences are not only between China and BRI economies, but also within BRI economies. Thus, these transnational differences are a set of differences between many stakeholders.

In general, the business cycle synchronization directly reflects the co-movement of output growth and fluctuations of various countries, and reveals the coordination and difference of transnational economic policies and the policy coordination of various countries to deal with shocks [4]. There are abundant researches on business cycle synchronization, but most of them focus on its measurement and determinants [5–10]. Through comparison, it is found that these studies only focus on the business cycle synchronization between one country and the other, which cannot fully reveal the interaction synchronization among multiple countries. In fact, the real economic system has the basic characteristics of social network, and the overall economic activity is caused by the interaction of many separated and different economic subjects, including division of labor, cooperation, trade and other multiple interactions [11]. Therefore, revealing the complex interdependence between economies is key to understanding the business cycle synchronization [12].

However, in the real quantitative analysis, the traditional statistic analysis method cannot reveal the complex linkages of business cycles synchronization. Fortunately, the complex network analysis method investigates and studies economic and social phenomena and structures from the perspective of "linkages", and can accurately reveal some intricate social and economic linkages between nodes such as countries [13, 14]. At present, complex network analysis methods are widely used in some academic subjects such as international trade [15], international investment [16], finance [17], macroeconomic volatility [18], industrial economy [19] and regional economy [20].

With the in-depth application of network analysis methods, complex network analysis has become an emerging trend to study the business cycles synchronization between countries [21, 22], and it is also an ideal tool to reveal the topological characteristics of the BCSN [23, 24]. Base on the concept of graph theory and complex network theory, countries can be regarded as nodes, the business cycle synchronization linkages of each country can be regarded as the connecting edges of nodes, and thus the BCSN is formed [21]. In empirical studies, pairwise correlation coefficient matrices can be used to represent the business cycle synchronization between different economies, and the topological structure of the BCSN can be studied by using complex network analysis method [12, 25].

Through further literature sorting, there were some studies analyzed the business cycle synchronization between China and other BRI economies [4, 25], and investigated the topological characteristics of BCSN between China and other ASEAN member states under the BRI [26]. Although these studies provide some reference for this paper, the research on the topological characteristics of the BCSN of BRI economies is still not profound enough. Then, what are the topological features of the BCSN of BRI economies? How does China fit into the network? At the same time, what are the differences in the topological features of the BCSN of BRI economies before and after the implementation of the BRI?

In order to answer all these questions, this paper uses the instantaneous quasi-correlation method to measure the level of business cycle synchronization for 53 BRI sample countries in

2000–2019, and construct pairwise business cycle synchronization matrix, and then uses complex network analysis method to construct the BCSN, and empirically studies the whole and individual topological characteristics of the BCSN in order to enrich the existing research. In the end, this paper will reveal China's influence in the network and compare the differences in the topological characteristics of the BCSN before and after the BRI was implemented. Based on the above synchronization matrix and network analysis method, this paper can find the characteristics of heterogeneity and diversity of the output synchronization linkages among various economies, revealing the interdependence and interactions of output linkages [22] and provide a new research perspective for the study of BRI business cycle synchronization.

The rest of this paper is arranged as follows. Section two shows a literature review related to this paper. Section three illustrates the methodology explanation and data description. Section four introduces the overall topological characteristics of BCSN. Section five reveals the individual topological characteristics of BCSN. And the last section is conclusions and discussions.

## 2. Related literature

The literature closely related to this paper mainly includes the following two categories: one is the measurement and research on the international business cycle synchronization; the other is on the BCSN.

The measurement and research on international business cycle synchronization mainly includes static method and dynamic method. For the static methods, existing studies mainly adopt simple correlation coefficient method [21, 27] and dynamic factor model [28, 29]. It should be noted that although the static method can intuitively judge the level of the synchronization, it cannot reveal the dynamic characteristics of the business cycle synchronization. Therefore, the follow-up measurement research has gradually shifted from the traditional static methods to the dynamic methods. For dynamic methods, existing studies mainly focus on Markov regime switching model [30–32], GARCH model [33, 34], concordance index [35], difference method [36, 37] and instantaneous quasi-correlation method [38, 39]. In addition, some scholars used the Method of Hodrick-Prescott (HP) filter to de-trend the original output data series and further calculate the business cycle synchronization [40, 41]. However, Hamilton (2018) believes that HP filtering method introduces unreal dynamic linkages that are not based on original data, and its results are affected by the size of smoothing parameter, so it cannot truly reflect the level of business cycle synchronization [42]. Compared with other dynamic methods, the instantaneous quasi-correlation method is more efficient and could carry out dynamic calculation and analysis, which avoids problems caused by artificial parameter setting and distortion of original data. Therefore, it has been widely used in practical calculation and analysis [38, 43].

As for the research on the BCSN, the existing research is mainly conducted from the perspective of network analysis. Diebold and Yilmaz (2013) investigated the linkages between actual outputs of G7 countries from 1962 to 2010 by network analysis, and found that the indicator of density could measure the pairwise output fluctuation linkages between different countries. At the same time, global connectedness will change as the business cycle changes [17]. Gomez et al. (2013) adopted correlation coefficient matrix and network analysis method to systematically investigate the co-movement of business cycles synchronization in various countries since 1950, and believed that the dynamic changes of interdependence among countries was mainly driven by the co-movement of regional economic growth rather than co-movement of world economy [12]. Caraiani (2013) found that compared with the Granger causality method, using correlation coefficients to construct a directed business cycle synchronization network can reflect the relative influence of countries in the world economy more

reasonably, and further empirical findings showed that the United States finally occupies the core position of the business cycle synchronization network of G7 and OECD [44]. Papadimitriou et al. (2014) used Pearson correlation coefficient and minimum dominating set to make an empirical study on the BCSN structure of 22 EU sample member states, and found that after the adoption of the common currency euro, the output of member states had a higher correlation, and the BCSN density of EU was increasing [45]. Xi et al. (2014) constructed the BCSN of G7 based on the pairwise maximum entropy model, and found that the network presented a clustering hierarchy and nearly accounted for almost half of the entire structure of the interactions within the G7 system [25]. Antonakakis et al. (2015) used sign concordance index and threshold-minimum dominating set method to investigate topological characteristics of BCSN among 27 countries during 1875–2013 and find that there are obvious differences in node degree of different countries in different periods [46]. Gomez et al. (2017) used correlation coefficient and minimum spanning tree technique to construct the BCSN of EU, analyzed the business synchronization linkages and accessibility among member states, and found that there was no obvious core-periphery structure inside the network [22]. Ductor and Leva-Leon (2016) adopted the social network analysis method and the indicator of betweenness centrality to evaluate the relative influence of various countries for global BCSN [32]. It is found that a country's is more influential in the network tends to increase when the economy is in recession, but becomes less influential when the economy is expanding. With Pearson correlation coefficient, rolling window, and threshold-minimum dominating set methods, Papadimitriou et al. (2016) selected some indicators such as the total number of edges, network density, the number of dominant and isolated nodes and node degree to empirically investigate the topological characteristics of European BCSN during 1986–2011 [45]. Based on the dynamic network analysis, Matesanz and Ortega (2016) used similarity index and minimum spanning tree (MST) technique to construct the European BCSN from 1950 to 2013, and found that the correlations and connectivity of the network increased significantly in 2009 [47]. With the help of correlation coefficient index used by Cerqueira (2013) [9], Belke et al. (2017) investigated the business cycle synchronization of the European Monetary Union, focusing on the core-periphery mode of the business cycle within the Union after the economic crisis [48]. Leiva-Leon (2017) established American inter-state BCSN by Markov Regime Switching framework, and investigated the its evolution model with indicators of multidimensional scaling (MDS) and closeness centrality, and found that the network has an obvious core-periphery structure [31]. Sebestyén and Iloskics (2020) employed the pairwise Granger causality between national outputs to construct the global shock contagion network, and found that it has a relative long path length and stronger transmission, and the degree distribution tends to be asymmetric [24].

For the research on the BSCN of BRI economies, Huang and Yao (2018) used CM (Cerqueira & Martins) synchronization index to measure the business cycle synchronization between China and BRI economies and find that it shows a certain "decoupling" trend, and there are obvious differences in different stages and different development aspects [4]. Cui et al. (2020) used dynamic correlation coefficient to calculate and found that China and Southeast and Central Asian countries, as well as Mongolia, Nepal, Pakistan, Sri Lanka and other countries have a high business cycle synchronization level [49]. Du et al. (2020) found that the BRI strengthened the business cycle synchronization linkages between China and ASEAN countries, while the network density and clustering coefficient also increased to some extent [26].

Based on all the related literature above, the existing researches have made great progress in the measurement and of topological characteristics BCSN, which provides theoretical and methodological support for this study. However, the research on the BCSN of BRI is still in the preliminary exploratory stage, and its topological characteristics have not been clearly

explained. Therefore, compared with previous studies, this paper may have three main contributions. Firstly, this paper systematically investigates the business cycle synchronization linkages among BRI economies by using the instantaneous quasi-correlation method, so as to reveal the relevant stylized facts. Different from existing studies, Huang and Yao (2018) and Cui et al. (2020) only analyzed the unidimensional business cycle synchronization between China and other BRI economies, but did not further reveal the pairwise multidimensional synchronization linkages among other countries [4, 49]. Secondly, 53 sample countries were selected to fully reveal the business cycle synchronization linkages among the BRI economies. Different from Du et al. (2020) [26], which only studies the local BCSN between China and ASEAN, this paper selects more sample economies for analysis, which can enrich the existing researches. Thirdly, this paper not only analyzes the topological characteristics of BCSN from the perspectives of overall and individual characteristics, but also compares the topological characteristics of the two phases before (2000–2013) and after (2014–2019) the BRI, in order to reflect the impact of the BRI on the structural evolution of BCSN. To sum up, based on pairwise correlation matrix and network analysis method [50, 51], this paper can better analyze the business cycle synchronization of BRI economies, and fully reveal the topological characteristics of the BCSN.

## 3. Research methods and data description

### 3.1 Measurement of business cycle synchronization

According to Duval et al. (2016) [39], this paper uses instantaneous quasi-correlation method to measure the BRI business cycle synchronization. The advantage of this method is that it does not adhere to the limitation of the range of the traditional correlation coefficient -1 to 1 and considers calculation mode of difference method. Besides, it is needless to consider the window period and the selection of filtering method, which can be used for dynamic calculation analysis, and thus is widely used [38, 43]. The calculation formula of this method is as follows:

$$S_{ijt} = \frac{(y_{it} - \bar{y}_i)\left(y_{jt} - \bar{y}_j\right)}{\theta_i \theta_j} \tag{1}$$

Where $y_{it}$ and $y_{jt}$ is the real GDP growth rate of country $i$ and country $j$ in year $t$, and $\bar{y}_i$ and $\bar{y}_j$ is the mean of real GDP growth rate, $\theta_i$ and $\theta_j$ is the standard deviation of real GDP growth rate of the two countries, and the real GDP growth rate is measured by the logarithmic difference method. Eq (1) could measure the inter-temporal business cycle synchronization between the two economies in the BRI, and finally form a symmetric instantaneous quasi-correlation coefficient matrix.

### 3.2 Complex network analysis methods

BCSN Construction. From the perspective of complex network, the network is a set composed of multiple nodes (social actors) and edges between nodes (linkages between actors) [14]. In fact, the international BCSN is a complex economic network, not just a simple collection of multiple economies, but also the linkages of business circle synchronization among them should be considered.

According to complex network theory and graph theory, BRI economies are regarded as nodes, and the business cycle synchronization (i.e. the value of instantaneous quasi-correlation between economies are regarded as edges between nodes). Therefore, according to Antonakakis et al. (2015) and Papadimitriou et al. (2016), the BCSN of BRI could be expressed as:

$G = \{V, E\}$, where $V = \{v_1, v_2, \ldots, v_n\}$ represents the node set composed of the BRI economies, $E = \{e_1, e_2, \ldots, e_n\}$ represents edges set composed of business cycle synchronization of BRI economies [45, 46]. Taken into account of the symmetry characteristics of the instantaneous quasi-correlation matrix, a undirected and unweighted BCSN composed of $N$ nodes can be constructed, in which the number of nodes $N$ is 53 sample economies.

In order to do the network structure analysis, the mean value $(\sum_i^N \sum_j^N S_{ij}/N(N-1))$ of the weighted synchronization matrix is usually taken as the threshold value. The above undirected and weighted BCSN could be unweighted to obtain the adjacency matrix $A$ composed of 0 and 1. Specifically, the mean of the undirected and unweighted instantaneous quasi-correlation matrix is firstly calculated, and then the value of the instantaneous quasi-correlation and the mean between different economies in the matrix is compared. Finally, if the value greater than the mean is set to 1, and the actual effective synchronization linkage is indicated; if the value is set to 0, representing the invalid synchronization linkage. At this point, the non-diagonal elements $a_{i,j}$ ($i \neq j$) of adjacency matrix $A$ can be defined as:

$$a_{i,j} = \begin{cases} 1, & \text{if } S_{ij} \geq \sum_i^N \sum_j^N S_{ij}/N(N-1) \\ 0, & \text{if } S_{ij} < \sum_i^N \sum_j^N S_{ij}/N(N-1) \end{cases} \tag{2}$$

Therefore, in an undirected and unweighted BCSN, the edges between different economies (nodes) reflects the validity of the business cycle synchronization linkage. For the same node ($i = j$), the value of the main diagonal in the corresponding matrix is zero. Next, by referring to Qiu and Liu (2021) [52], this paper investigates the topological characteristics of the BCSN of BRI economies from the overall and individual sides.

Indicators of overall characteristics of network structure. In this paper, indicators such as density, network efficiency, clustering coefficient, average path length and condensed subgroup are selected to analyze the overall topological characteristics of BCSN evolution of the BRI economies.

Network density (DS) describes the degree of business cycle synchronization between the BRI economies, which can be calculated by the ratio of the actual effective linkage number $M$ to the theoretical maximum linkage $N(N-1)$ number of the BCSN in year $t$. At this point, the network density can be calculated by $DS_t = 2M_t/N(N-1)$.

Network efficiency (NE) reflects the accessibility of economic fluctuation correlation among the BRI economies, and is usually expressed as the reciprocal of distance between all nodes in year $t$, that is $h_{ijt}$. The corresponding formula is $NE_t = \sum_{i=1}^N \sum_{j=1}^N h_{ijt}/N(N-1)$.

Clustering coefficient (CC) reflects the overall degree of interconnection among all economies in the network and their close neighbors, so as to describe the degree of clustering among some nodes [15]. When the degree of the node i in the year $t$ is $D_{it}$, and the number of edges between it and all its neighboring nodes is $E_{it}$, so the formula of the CC is $CC_t = E_{it}/N[D_{it}(D_{it}-1)]$.

Average Path Length (AL) represents the mean of connected edges that the shortest path length between all potentially connected nodes in the network passes through [24]. It reflects transmission efficiency of the business cycle synchronization linkage between nodes. Assuming that the shortest path length between node i and j in the network is $d_{ijt}$, the formula is $AD_t = \Sigma_i \Sigma_j d_{ijt}/(N^3 - N) - 1/N$.

The Quadratic Assignment Procedure (QAP) correlation analysis is a non-parametric test method for calculating the correlations between matrices of different variables based on random matrix permutation [13, 53]. By analyzing the correlation and significance level of business cycle synchronization matrix at different times, it can reflect the dynamic evolutionary characteristic of network structures.

The cohesive subgroups reflect the composition of subgroups and the tightness of node linkages in the network, which is correspondent to subgroup density matrix reflecting the tightness of the business cycle synchronization correlation between each subgroup and its external subgroup. According to the block model theory, network roles can be generally divided into four categories: main beneficiary subgroup, net spillover subgroup, broker subgroup and two-way spillover subgroup [54].

Indicators of individual characteristics of network structure. In this paper, indicators such as eigenvector centrality, betweenness centrality, coreness are selected to analyze the individual topological characteristics of BCSN evolution of the BRI economies.

Eigenvector centrality (EC) reflects the relative influence of nodes in the BCSN. It considers structure type of the network, which represents the weighted average sum of all direct and undirected connections of each node, that is, its value is affected by the centrality of neighboring nodes [55].

Betweenness centrality (BC) measures the controlling ability of a node over the BCSN, that is to describe the "hub" role played by an economy in transmitting economic fluctuation influence in the network [31]. Assuming that the number of shortest paths between node j and k in the network at year t is $\bar{N}_{jkt}$, and the corresponding total number of shortest paths is $N_{jkt}$, so the formula is $BC_{it} = \sum_{j \neq k; j, k \neq i} \bar{N}_{jkt} / N_{jkt}$.

Coreness (COR) reflects the core status of a node in the BCSN, and reveals the special structure of the business cycle synchronization between the core and peripheral nodes in the network through core-periphery analysis.

## 3.3 Data description and source

To the choice of sample, we refer to the standards from China's BRI website (https://eng.yidaiyilu.gov.cn/index.htm). Comprehensively considering the availability and consistency of data, 53 sample BRI economies from 2000 to 2019 are selected as the research objects in this paper. More details of 53 sample BRI economies are available in S1 Appendix. The real GDP data used in this paper is expressed as GDP constant 2010 US$, and the data comes from the World Development Indicators (WDI) database in the World Bank. Based on Eqs (1) and (2), we collate the matrices data of undirected and unweighted quasi-correlation coefficient for different period. Details of the relevant data can be found in S1 Data.

## 4. BCSN structure: Overall characteristics analysis

### 4.1 Analysis of network structure evolution

As can be seen from Table 1, the network density and network efficiency of the BCSN from 2000 to 2019 have obvious stage characteristics, showing a fluctuating downward trend, indicating that the business cycle synchronization linkages between economies is still in a relative weak connection state. From the perspective of different periods, compared with 2000–2013,

**Table 1. Statistical results of overall characteristics.**

| Indicators | 2000 | 2002 | 2004 | 2006 | 2008 | 2010 | 2012 | 2014 | 2016 | 2018 | 2019 | 2000–2013 | 2014–2019 |
|---|---|---|---|---|---|---|---|---|---|---|---|---|---|
| DS | 0.470 | 0.495 | 0.477 | 0.462 | 0.504 | 0.495 | 0.395 | 0.388 | 0.382 | 0.532 | 0.427 | 0.493 | 0.429 |
| NE | 0.493 | 0.495 | 0.535 | 0.573 | 0.495 | 0.495 | 0.429 | 0.441 | 0.434 | 0.752 | 0.462 | 0.734 | 0.700 |
| CC | 0.964 | 0.999 | 0.889 | 0.922 | 1.000 | 1.000 | 0.927 | 0.913 | 0.892 | 0.942 | 0.959 | 0.750 | 0.737 |
| AL | 1.057 | 1.001 | 1.290 | 1.406 | 1.000 | 1.000 | 1.142 | 1.227 | 1.253 | 1.560 | 1.150 | 1.487 | 1.659 |

Note: The results in the table were collated according to the Cohesion algorithm under the NETWORK module of Ucinet6 software.

the value of DS and NE decreased during 2014–2019. Therefore, since the inception of the BRI in 2013, the degree of output synchronization linkages and the influence of corresponding economic linkages of BRI economies is weakened.

It is shown that the real and effective synchronization relationship between countries involving in BRI from 2000 to 2012 was obviously adversely affected by the 2008 global financial crisis, and the accessibility of economic fluctuation and risk correlation was also rapidly declining, which also indicated that countries adopted relatively active anti-business cycle prevention policies to resist the impact of the financial crisis. After the BRI was put forward in 2013, the closeness of BCSN has been strengthened, and the accessibility of corresponding economic links has also been improved, indicating that BRI aimed at building a mutually beneficial and win-win "community of shared future" had a positive impact on enhancing economic and trade cooperation and links among the BRI economics.

Furthermore, the clustering coefficient decreased on the whole, and reached its peak during 2008–2010, indicating that the synchronization linkages among some economies inside the network had obvious clustering characteristics. Meanwhile, the shock of the global financial crisis in 2008 made the clustering characteristic more obvious. In addition, the average path length increased slightly on the whole, proving that the path length required for the transmission of the synchronization linkage is relatively short, that is, the influence of economic output and fluctuations of any country in the network only need to be transmitted once to reach other countries. In particular, compared with 2000–2013, the mean value of clustering coefficient during 2014–2019 decreased slightly, while the mean path length increased significantly, indicating that the clustering degree of output synchronization was weakened after the inception of the BRI. Further analysis revealed that the BCSN of the BRI economies always show a large clustering coefficient and a small average path length, which has the typical "small world" topological characteristic [56, 57].

From Table 2, correlation coefficient of business cycle synchronization matrix of BRI economies are basically near zero value, and most of them fail the significance test in 2000–2013, showing obvious weak related or unrelated characteristics. Hence, the BCSN of BRI economies does not have typical evolutionary characteristics of self-stability before the inception of the

**Table 2. QAP correlation result.**

| QAP | 2000 | 2002 | 2004 | 2006 | 2008 | 2010 | 2012 | 2014 | 2016 | 2018 | 2019 | 2000–2013 | 2014–2019 |
|---|---|---|---|---|---|---|---|---|---|---|---|---|---|
| 2000 | 1.000 | | | | | | | | | | | | |
| 2002 | -0.014 | 1.000 | | | | | | | | | | | |
| 2004 | 0.065 | -0.004 | 1.000 | | | | | | | | | | |
| 2006 | 0.032 | 0.007 | -0.048 | 1.000 | | | | | | | | | |
| 2008 | 0.005 | -0.018 | -0.017 | -0.012 | 1.000 | | | | | | | | |
| 2010 | 0.032 | -0.018 | -0.017 | -0.018 | -0.010 | 1.000 | | | | | | | |
| 2012 | 0.035 | 0.033* | 0.076 | 0.162** | -0.004 | 0.041* | 1.000 | | | | | | |
| 2014 | 0.052 | 0.06** | 0.053 | 0.184** | -0.013 | 0.005 | 0.104** | 1.000 | | | | | |
| 2016 | 0.027 | 0.012 | -0.043 | 0.081 | -0.016 | -0.010 | -0.144** | 0.111* | 1.000 | | | | |
| 2018 | -0.025 | -0.012 | -0.039 | -0.007 | 0.004 | 0.033 | -0.021 | 0.017 | 0.097*** | 1.000 | | | |
| 2019 | 0.001 | -0.024 | -0.002 | 0.013 | -0.025 | 0.022 | -0.032 | 0.054 | 0.236*** | 0.298*** | 1.000 | | |
| 2000–2013 | | | | | | | | | | | | 1.000 | |
| 2014–2019 | | | | | | | | | | | | 0.060 | 1.000 |

Note: The results of this table are based on the QAP correlation algorithm in testing hypotheses.

*, ** and *** represent significance levels of 10%, 5% and 1% respectively.

**Table 3. Density matrix variation and group relations.**

| | Changes of density matrix | | | | Role of subgroups during 2014–2019 | | | | | | | |
|---|---|---|---|---|---|---|---|---|---|---|---|---|
| | S1 | S2 | S3 | S4 | S1 received | S2 received | S3 received | S4 received | Number of Nodes | Expected Relation Ratio | Actual Relation Ratio | Role |
| S1 | 0.816 (0.399) | (-0.201) | (0.088) | (-0.343) | 222 | 191 | 11 | 3 | 17 | 0.308 | 0.520 | Two-way Spillover group |
| S2 | 0.562 | 0.763 (-0.094) | (-0.202) | (0.105) | 191 | 290 | 1 | 52 | 20 | 0.365 | 0.543 | Broker group |
| S3 | 0.324 | 0.025 | 1.000 (0.100) | (-0.455) | 11 | 1 | 2 | 5 | 2 | 0.019 | 0.105 | Net Spillover group |
| S4 | 0.013 | 0.186 | 0.179 | 0.780 (-0.083) | 3 | 52 | 5 | 142 | 14 | 0.250 | 0.703 | Primary Benefit group |

Note: The table is based on the CONCOR algorithm in Roles & Positions under the NETWORK module of Ucinet6 software. The density matrix is symmetric, and the values in brackets are the changes of density values from 2014 to 2019 compared with 2000–2013. Expected relation ratio = (the number of nodes in the group minus one) / (the number of all nodes minus one), and actual relation ratio = the number of internal contacts in the subgroup/the total number of external links issued by the subgroup.

BRI. However, during 2014–2019, although the correlation of synchronization matrix is still low, it has gradually increased and passed the significance test. It is indicated that the positive correlation of synchronization matrix was dynamically strengthened, network structure gradually highlighted the progressive evolutionary characteristics as the proposal and practice of the BRI. Meanwhile, the correlation coefficient of the synchronization matrix during 2000–2013 and 2014–2019 is 0.060, which fails to pass the significance test. Therefore, there is no significant correlation for the BCSN of BRI economies before and after the inception of the BRI.

## 4.2 Structure analysis of network subgroup

In order to better reveal the subgroup structure changes of BCSN of the BRI economics before and after the inception of the BRI, Table 3 shows the change of density matrix of the BCSN and the role of subgroups during 2014–2019, and structure diagrams of the two network subgroups are obtained with the support of VOSviewer software (Figs 1 and 2). Furthermore, from the evolution of network structure, subgroup 3 and Subgroup 4 were the main parts of network structure during 2000–2003, while subgroup 1 (two-way overflow subgroup) and subgroup 2 (broker subgroup) were the main parts of network structure during 2014–2019. This shows that the BRI has strengthened the business cycle synchronization linkages of BRI economies, and formed a spreading correlated structure in space with China, Singapore, India, Saudi Arabia, Turkey, Russia and Kazakhstan as main nodes.

The number of nodes in subgroup 1 has increased, mainly including countries such as China, Singapore and Turkey, and two other isolated nodes (Nepal and Yemen). After the inception of the BRI, the synchronization density of nodes inside subgroup 1 was relatively large and had the largest increase. Secondly, the subgroup 1 connection degree with subgroup 3 increased, while the one with Subgroup 2 and subgroup 4 decreased significantly. The number of internal received relations of the subgroup 1 is 222, the number of external sent relations is 205, the corresponding expected relation ratio is 0.308, and the actual relation ratio is 0.520, indicating that the Subgroup 1 is a two-way spillover group.

Subgroup 2 mainly contains India, Israel, Russia and other countries, and the number of its internal nodes has increased significantly. In addition to the decline of synchronization degree of internal connection, the synchronization degree of external connection has increased significantly. The number of internal received and external sent relations of subgroup 2 is 290 and 244 respectively, and the expected relation ratio and actual relation ration is 0.365 and 0.543

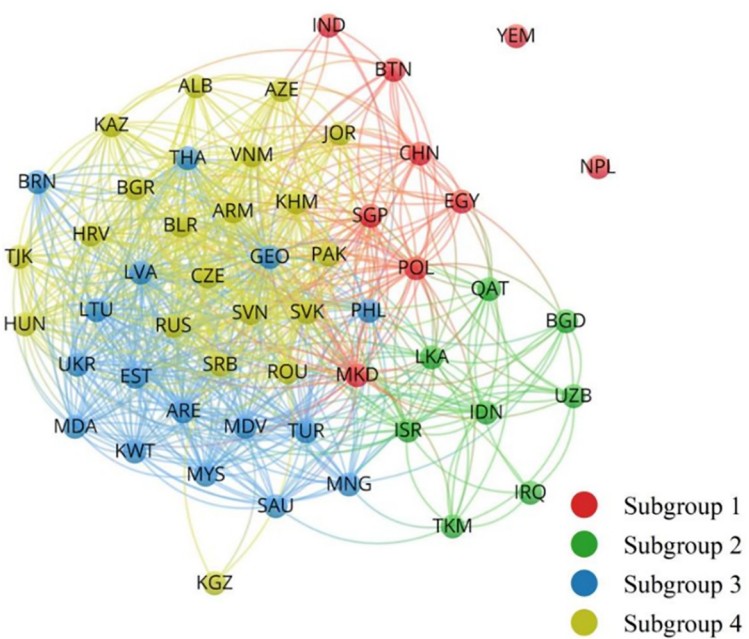

**Fig 1. Network subgroups during 2000–2013.** Source: Drawn by the authors from VOSviewer software.

respectively, which has typical broker group characteristics. It can be seen that after the inception of the BRI, the internal connection degree of subgroup 2 decreased, but its external connection degree increased significantly, and it took an important mediating effect in the whole network.

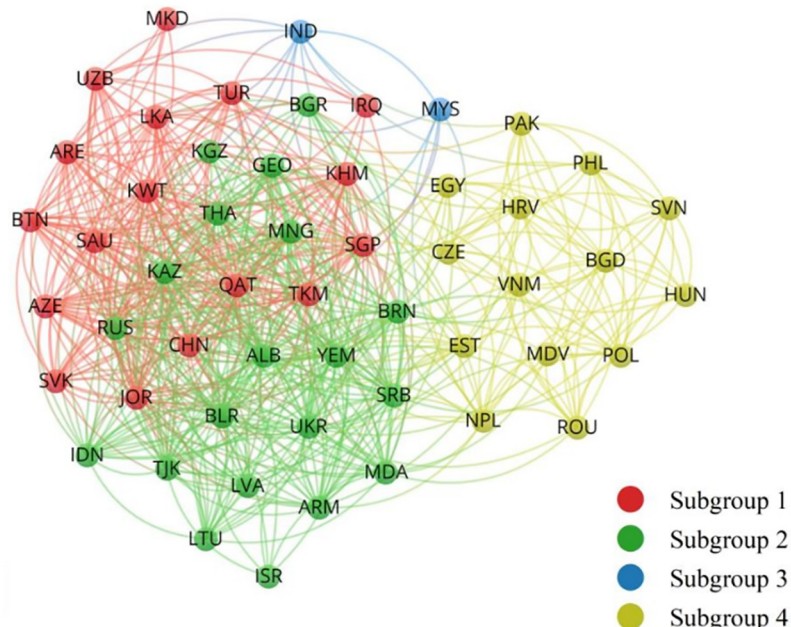

**Fig 2. Network subgroups during 2014–2019.** Source: Drawn by the authors from VOSviewer software.

The number of nodes inside subgroup 3 decreased significantly, and a relatively isolated subgroup containing only Indonesia and Malaysia was finally formed. At this time, synchronization linkages inside subgroup 3 became closer after the inception of the BRI. And the synchronization degree of subgroup 3 with subgroup 1 and subgroup 2 increased while the one with subgroup 4 decreased. The number of linkages received inside and sent out from subgroup 3 is 2 and 17 respectively and the expected relation ratio and actual relation ratio is 0.019 and 0.105, respectively, showing obvious outward spillover characteristics and belonging to the net spillover group.

The number of nodes inside subgroup 4 has decreased mainly including countries like Vietnam, the Philippines, Pakistan, Poland and Romania. Synchronization linkages inside subgroup 4 became looser after the inception of the BRI, while the linkage degree with other three subgroups increased. The number of linkages received inside and sent out from subgroup 4 is 142 and 60, and the expected relation ratio and actual relation ratio is 0.250 and 0.703, respectively. It is a typical beneficiary in the BCSN and belongs to the beneficiary group.

## 5. BCSN structure: Individual characteristics analysis

In order to better reveal the evolution of individual characteristics of BCSN structure before and after the inception of BRI, this section will mainly show the individual characteristic values and their changes and corresponding rankings in main BRI economies from 2014 to 2019 (Table 4), and separately analyze the results and rankings of China's individual characteristic in 2000–2019 (Table 5).

**Table 4. Statistics of individual characteristics of main economies during 2014–2019.**

| Country/Region | Eigenvectors Centrality | | | Betweenness Centrality | | | Node Coreness | | |
|---|---|---|---|---|---|---|---|---|---|
| | EC | Changes | Rank | BC | Changes | Rank | NC | Changes | Rank |
| China | 0.186 | 0.038 | 3 | 3.280 | -3.710 | 23 | 0.188 | 0.070 | 5 |
| Mongolia | 0.134 | 0.058 | 18 | 4.784 | 3.208 | 19 | 0.177 | 0.064 | 14 |
| Singapore | 0.182 | 0.054 | 10 | 1.256 | -0.021 | 40 | 0.182 | 0.044 | 7 |
| Thailand | 0.121 | 0.012 | 22 | 4.037 | 2.676 | 21 | 0.182 | 0.044 | 7 |
| India | 0.061 | -0.036 | 43 | 4.965 | 4.095 | 18 | 0.068 | 0.038 | 43 |
| Pakistan | 0.037 | -0.041 | 48 | 0.260 | -1.572 | 48 | 0.063 | -0.055 | 46 |
| Saudi Arabia | 0.122 | -0.001 | 21 | 1.757 | 0.176 | 32 | 0.148 | 0.040 | 24 |
| Egypt | 0.069 | -0.036 | 39 | 2.086 | -0.269 | 29 | 0.086 | 0.007 | 37 |
| Russia | 0.188 | 0.036 | 1 | 5.446 | 4.360 | 16 | 0.182 | -0.010 | 7 |
| Ukraine | 0.074 | -0.058 | 36 | 138.189 | 136.219 | 1 | 0.200 | 0.037 | 1 |
| Kazakhstan | 0.186 | 0.047 | 4 | 5.153 | 2.171 | 17 | 0.200 | 0.057 | 1 |
| Tajikistan | 0.185 | 0.092 | 7 | 44.393 | 41.606 | 3 | 0.177 | 0.049 | 14 |
| East Asia | 0.160 | 0.048 | 1 | 4.032 | -0.251 | 4 | 0.183 | 0.067 | 1 |
| Southeast Asia | 0.117 | 0.012 | 4 | 2.845 | 0.727 | 5 | 0.133 | -0.007 | 4 |
| South Asia | 0.067 | -0.021 | 6 | 1.765 | -1.126 | 6 | 0.101 | 0.031 | 6 |
| West Asia & North Africa | 0.128 | 0.022 | 3 | 6.314 | 3.359 | 3 | 0.139 | 0.024 | 2 |
| Central & Eastern Europe | 0.092 | -0.032 | 5 | 18.908 | 17.000 | 1 | 0.118 | -0.047 | 5 |
| Central Asia | 0.139 | 0.041 | 2 | 10.642 | 8.440 | 2 | 0.135 | 0.055 | 3 |

Note: The results are compiled from the Centrality and Power algorithms and Core/Periphery algorithms under the NETWORK module of Ucinet6 software. The change value was the difference between 2014–2019 and 2000–2013, and the ranking was compiled according to the results of 2014–2019. The regional result is the mean of the sample eigenvalues inside the region.

**Table 5. Statistics on individual characteristics of China in 2000–2019.**

| Indicator | 2000 | 2002 | 2004 | 2006 | 2008 | 2010 | 2012 | 2014 | 2016 | 2018 | 2019 |
|---|---|---|---|---|---|---|---|---|---|---|---|
| EC | 0.000 | 0.180 | 0.088 | 0.184 | 0.186 | 0.186 | 0.182 | 0.190 | 0.194 | 0.170 | 0.184 |
| | (34) | (28) | (36) | (4) | (1) | (1) | (20) | (7) | (7) | (21) | (1) |
| BC | 0.000 | 0.000 | 0.000 | 43.208 | 0.000 | 0.186 | 0.000 | 5.931 | 6.921 | 0.000 | 0.160 |
| | (13) | (28) | (27) | (4) | (10) | (1) | (10) | (7) | (7) | (31) | (13) |
| NC | 0.067 | 0.144 | 0.075 | 0.205 | 0.149 | 0.149 | 0.169 | 0.191 | 0.198 | 0.141 | 0.168 |
| | (46) | (28) | (36) | (4) | (1) | (1) | (20) | (7) | (7) | (23) | (1) |

Note: Calculated by the authors from UCINET6. The values in brackets are the rankings.

### 5.1 Node centrality analysis

Table 4 shows that, on the whole, the value of EC and BC of main BRI economies are still low. There are obvious transnational differences within different regions. This shows that the influence of main BRI economies in the BCSN still needs to be improved, and they have not effectively played the role of "hub". After comparison, it is found that after the inception of the BRI, the relative influence of East Asia and Central Asia in the network is more prominent, while central and Eastern Europe and Central Asia make a better mediating effect. At the same time, some economies, such as China, Russia, Ukraine, Kazakhstan and Tajikistan, do not match the influence and mediating effect in the network, but perform better than other economies as a whole.

For different geographic regions, EC of East Asian ranked first, but its CC ranked fourth. Among it, China's EC improved while CC had an obvious decrease. It shows that East Asia has a certain influence, but the core hub role is not prominent. Further combined with Table 5, during the sample period, China's relative influence in the network has significantly increased, and it has more influence after the inception of the BRI, while its CC is relatively low.

Although the overall level of centrality in Southeast Asia has improved, its ranking is still relatively low. Singapore's relative influence is stronger than Thailand's, but Thailand's hub role is more obvious. Both EC and CC of South Asia declined and ranked the last. Meanwhile, the centrality levels of Pakistan and India inside the region were low, which did not exert a certain influence and mediating effect on the regional and external business synchronization. Central Asia, which is deeply inland and geographically close to China, has the second highest level of centrality, in which Kazakhstan has a higher relative influence and Tajikistan plays a more prominent mediating role.

The centrality levels of West Asia and North Africa have improved, while the relative influence of the node countries Saudi Arabia and Egypt has declined, and they have not played an obvious hub role. There are significant differences in the EC and CC of Central and Eastern Europe (CEE), with their relative influence reduced and far inferior to their hub role and there are also some significant differences in node countries inside the CEE.

### 5.2 Node coreness analysis

It can be seen from Table 4 that, on the whole, the BCSN of BRI economies is characterized by the coexistence of "multi-core" and "multi-periphery" and consist of multiple levels of cores, semi cores, and peripheries. Similar to the results of node centrality, there are significant transnational differences in coreness degree within different regions. Specifically, East Asia, West Asia and North Africa have a higher coreness level and occupy the relative core position of the

network, while Central Asia and Southeast Asia are in the intermediate zone between the core and the periphery of the network, CEE and South Asia are in the periphery of the network.

Combined with Table 5, it can be seen that China's coreness level fluctuated from 0.068 to 0.168 during the sample period. The corresponding ranking reached first, which was similar to the EC calculated in the last section. It is shown that after the inception of the BRI, China is accelerating its integration into the BCSN which has a greater impact on output changes in other economies, and eventually occupies the core position of the network. This may be due to China's strong driving force for economic growth and higher level of opening-up policy, as well as its systematic and reliable economic and financial risk prevention policy, and the formation of a good and close relationship of coordinated economic development with the BRI economies.

From the perspective of different regions, although the coreness level of Mongolia is obviously worse than that of China, the gap between the two is small, contributing to the top ranking of East Asia as a whole. Singapore and Thailand achieved synchronized growth in coreness and tied for seventh place, but other countries in the region did not achieve high coreness level, resulting in the overall lagging behind. In South Asia, Pakistan and India are significantly lower in the coreness level and ranking, and at the periphery of the network. The coreness level of West Asia and North Africa has been improved and ranks second, but the core influence of Egypt and Saudi Arabia in the regional and external business synchronization linkage is not prominent. CEE ranked last in terms of coreness level, with Ukraine significantly higher than Russia. Central Asia has improved its coreness level, and Kazakhstan's core position is obviously better than Tajikistan.

## 6. Conclusion and discussion

With the support of instantaneous quasi-correlation and complex network analysis method, this paper empirically analyzed the topological characteristics of BCSN of BRI economies from 2000 to 2019. The main research conclusions are as follows:

First, in the sample period, the business cycle synchronization linkage of BRI economies is still relative weak, the network density and network efficiency has decreased after the inception of the BRI.

Second, the BCSN of BRI economies always show a large clustering coefficient and a short average path length, which presents a typical structural characteristic of "small world". Meanwhile, the clustering degree of output synchronization linkage of BRI economies is weakened after the inception of the BRI.

Third, on the whole, the BCSN of BRI economies does not have the characteristic of gradual evolution. But since the inception of the BRI, the evolution of the BCSN of BRI economies shows a self-stability characteristic.

Fourth, the BCSN structure is composed of four subgroups. The synchronization linkage level inside subgroups is obviously higher than the one outside subgroups. After the inception of the BRI, the BCSN structure of BRI economies mainly consist of two-way spillover subgroup and broker subgroup.

Fifth, the individual characteristics of the BRI economies are obviously different, the relative influence of a country in the network does not fully show its hub role. After the inception of the BRI, the function of China and other important nodes, such as Southeast Asia, Central Asia and CEE, has been enhanced. From the perspective of different regions, East Asia plays a relatively big role in the network, CEE and Central Asia take the most prominent mediating effect.

Sixth, China's EC and coreness level have increased and ranked top during the sample period, but its CC level is still low. After the inception of the BRI, China's relative influence in the entire network has increased significantly, but does not show much mediating effect.

Combined with instantaneous quasi-correlation and complex network analysis method, this paper reveals the structural characteristics of the BCSN of BRI economies from the overall and individual aspects. It should be noted that there are still some deficiencies in this paper, which need to be improved by follow-up research. Specifically, future research can be improved from the following three aspects. First of all, besides the instantaneous quasi-correlation method, other methods can be used to measure the business cycle synchronization, such as Markov switching model, dynamic correlation coefficient method, GARCH Model, concordance index and difference method, to ensure the robustness of empirical results. In addition, combined with cutting-edge complex network analysis methods, it is considered to make a weighted business cycle synchronization matrix, and further investigate the characteristics of the BCSN of BRI economies such as robustness and vulnerability, so as to enrich the existing research. Finally, Multiple Regression Quadratic Assignment Procedure (MRQAP) could be used to empirically study the driving factors of the structure evolution of the BCSN of BRI economies, which may provide some policy suggestions for strengthening the business cycle synchronization linkage of BRI economies.

## Supporting information

**S1 Appendix. Sample BRI economies.** This document contains additional details on the 53 sample BRI economies.
(PDF)

**S1 Data. Data tables.** These data sheets contain the original data on GDP constant 2010 US$, and the matrices data of undirected and unweighted quasi-correlation coefficient for the period of 2000–2013 and 2014–2019.
(XLSX)

## Acknowledgments

We are grateful to anonymous reviewers for constructive and insightful comments. All remaining errors are the responsibility of the authors alone.

## Author Contributions

**Conceptualization:** Zhiping Qiu.

**Data curation:** Zhiping Qiu.

**Formal analysis:** Zhiping Qiu, Sichao Mai.

**Methodology:** Zhiping Qiu.

**Software:** Zhiping Qiu.

**Visualization:** Zhiping Qiu.

**Writing – original draft:** Zhiping Qiu, Sichao Mai.

**Writing – review & editing:** Zhiping Qiu, Sichao Mai.

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
