## [Decision Letter · Decision Letter 0]

4 Apr 2022

PONE-D-21-31756

Topological Characteristics of International Business Cycle Synchronization: A Network Analysis of the BRI Economies

PLOS ONE

Dear Dr. Sichao,

Thank you for submitting your manuscript to PLOS ONE. After careful consideration, we feel that it has merit but does not fully meet PLOS ONE’s publication criteria as it currently stands. Therefore, we invite you to submit a revised version of the manuscript that addresses the points raised during the review process.

Two reviewers have evaluated your manuscript and provided the comments below (and in the attached PDF copy of the manuscript). While the reviewers indicated interest in the topic, they raised a number of significant issues that need to be addressed in order for the manuscript to be suitable for publication. Please revise to address all comments. The revised manuscript will be re-evaluated by these reviewers, if they are available.

We look forward to receiving your revised manuscript.

Kind regards,

Takashi Nishikawa, Ph.D.

Academic Editor

PLOS ONE

https://journals.plos.org/plosone/s/file?id=ba62/PLOSOne_formatting_sample_title_authors_affiliations.pdf".

Reviewers' comments:

Reviewer's Responses to Questions

**Comments to the Author**

1. Is the manuscript technically sound, and do the data support the conclusions?

Reviewer #1: Partly

Reviewer #2: Yes

2. Has the statistical analysis been performed appropriately and rigorously? 

Reviewer #1: No

Reviewer #2: I Don't Know

3. Have the authors made all data underlying the findings in their manuscript fully available?

Reviewer #1: Yes

Reviewer #2: Yes

4. Is the manuscript presented in an intelligible fashion and written in standard English?

Reviewer #1: No

Reviewer #2: No

5. Review Comments to the Author

Reviewer #1: Please see my detailed comments in your PDF manuscript as attached.

- Recalling the inception of the BRI in 2013, data analysis should be made:

- Remove redundant description of the BRI.

- Check several typos. Change "BRI economics" to BRI economies.

- Implcation should be rewritten.

Reviewer #2: General comments

The paper is interesting, its structure is correct, and the topic is clearly a hot one. Results are fine unless not surprising. However, from my point of view, the following main issues make its publication not recommended at its current state.

1. Introduction is extremely long and not very informative. At the end of it, we still do not know what the “BRI” is (who is financing the initiative, what for, what the achievements are, what kind of projects have been implemented, and so on)

2. The paper, from my point of view, is very poorly written and difficult to understand. There are plenty of examples, I will show just a couple of them:

• Page 8, “With the continuous advancement of economic globalization and international division of labor, the correlation trend of output periodicity and economic scale among countries presents a certain nonlinear and multi-threaded complex relationship, and then forms an interactive BCSN in space”

• Page 9, “are regarded as nodes in this paper, and the quasi-correlation coefficients between the actual economic scales of economics are regarded as the connecting edges between nodes”

3. Figures throughout the paper are plotted in an extremely low quality and they are no informative

4. The paper shows an important number of “speculative” statements, especially when it comes to policy advice. For instance, page 13, the authors claim: “It is worth noting that in 2019, network density and efficiency saw a significant decline, which was mainly caused by the unilateral trade protectionism and continuous trade frictions among western countries led by the US in recent years, which worsened the previous positive global economic and trade environment”. However, the authors do not analyse the variables affecting output synchronicity and consequently they cannot state such connection. In page 4 the authors claim that their paper will provide policy inspiration to enhance economic synchronicity among BRI countries, however this is not possible with the analysis the authors provide.

5. References regarding network analysis applied to synchronization is quite poor. The authors just cite 3 or 4 papers dealing with this issue but many more can be found and should be referenced in the paper. In the same line, the authors have not deeply revised the economic literature on business cycle synchronization.

6. I do not understand Table 1. In it the Networks density measure is provided. This measure is shown for different years. However, if I am correct, the edges of the network represent correlation degree while nodes represent countries. Therefore, what is the meaning of this measure for year 2000, which is the first year of analysis? Or the same measure for 2019? This is not clear to me

6. PLOS authors have the option to publish the peer review history of their article (what does this mean?). If published, this will include your full peer review and any attached files.

Reviewer #1: No

Reviewer #2: No

---

## [Author Response · Author response to Decision Letter 0]

13 May 2022

Responses to Reviewer #1: 

Dear reviewer:

Thank you for your comments on our manuscript. Those comments are valuable and very helpful for revising and improving our paper. We have studied the comments carefully and have made corrections which we hope to meet with approval finally.

Below we address each issue the reviewers raised and describe corresponding changes in the revised manuscript. For ease of reference, changes were made by using blue text in the revised manuscript. Reviewers’ comments are repeated in full using italics, while our responses are typed in standard.

1.General Comment: Please see my detailed comments in your PDF manuscript as attached.

Response: Thank you for your valuable time and efforts in this review process, and thanks for your excellent comments that could improve the quality of our manuscript. Following you detailed comments in PDF manuscript, we significantly improved the quality of this manuscript based on your constructive comments. Here we submitted a new version, which has been further revised according to your suggestions. In summary, major modifications include: 

First, we have carefully rewritten the abstract according to the research objectives, research methods, data, and the framework of the main findings  

Second, in the first introduction part, we introduced BRI in a more concise way, restated the research questions in this paper, and expounded the research content more accurately.  

Third, in the second part of related literature, we reorganized relevant literature, made some improvements in the measurement of business cycle synchronization, the research on the structural characteristic of the BCSN, and summarized the shortcomings of existing research and the contribution of this paper  

Fourth, in the third part of research methods and data description, we mainly improved the description of complex network analysis methods, especially the construction process of BCSN, and added the calculation formula of structural characteristic indicators.  

Fifth, in the fourth part of the overall characteristics analysis, we deleted the redundant results of weighted clustering coefficient (WCC), added measuring results and the analysis before and after the inception of the BRI, redrew the diagram of network subgroup structure with VOSviewer software, and elaborated overall structural characteristics of the BCSN from the network structure evolution and network subgroup structure. 

Sixth, in the part of individual characteristics analysis, we deleted the redundant node strength (NS) results, improved the interpretation of the measurement results before and after the inception of the BRI, and illustrated the individual structural characteristics of the BCSN from node centrality and coreness degree.  

Seventh, in the conclusion and discussion part, we rerefined the research conclusions, deleted the original policy recommendations, and added the discussion on future research prospects.  

Eighth, we tried our best to correct punctuation marks, misspellings, grammar, references and other details in the paper, and deleted the “speculative” statements, in order to improve the quality of this paper and meet relevant requirements.  

2.Major comments

Comment 1: Recalling the inception of the BRI in 2013, data analysis should be made.

Response 1: Thanks for your constructive comments. Following your suggestions, we had added the data analysis to recalling the inception of the BRI. The detailed revision is shown on pages 1-2 and below: 

The BRI comprises of over 50 different economies, which cover 80% of globe population and its the estimated cost is over $21.1 trillion US dollars (Klinger 2019). Since the BRI was proposed in 2013, major achievements have been made in infrastructure construction, international trade, cross-border investment and sustainable growth. One World Bank report[ For a more details of the achievements of the BRI, please refer to the World Bank report Belt and Road Economics:  Opportunities and Risks of Transport Communities, https://www.worldbank.org/en/topic/regional-integration/publication/belt-and-road-economics-opportunities-and-risks-of-transport-corridors.] noted that BRI is expected to increase real income by 1.2-3.4% in BRI economies and 0.7%-2.9% globally, with investments that could lift more than 7.6 million people out of extreme poverty. In particular, BRI transport projects have significantly reduced trade costs, which are expected to increase trade in BRI economies by 2.8% to 9.7%, increase in world trade by 1.7% to 6.2%, and total foreign direct investment in BRI economies by 4.97% (World Bank, 2019). 

Comment 2: Remove redundant description of the BRI.

Response 2: Thanks for your point. Following your suggestions and detailed comments in PDF manuscript, we had removed redundant description of the BRI in the second paragraph of the first page. The detailed revision is shown on pages 1-2 and below: 

The BRI was initiated by the Chinese president Xi Jinping in 2013, and it was completely introduced during the visit of Kazakhstan and Indonesia. In the background of the history of China's ancient overland and maritime Silk Road, the BRI consists of the Silk Road Economic Belt and the 21st Century Maritime Silk Road, geographically crossing Asia, Europe and Africa. The BRI comprises of over 50 different economies, which cover 80% of globe population and its the estimated cost is over $21.1 trillion US dollars (Klinger 2019). Since the BRI was proposed in 2013, major achievements have been made in infrastructure construction, international trade, cross-border investment and sustainable growth. One World Bank report[ For a more details of the achievements of the BRI, please refer to the World Bank report Belt and Road Economics:  Opportunities and Risks of Transport Communities, https://www.worldbank.org/en/topic/regional-integration/publication/belt-and-road-economics-opportunities-and-risks-of-transport-corridors.] noted that BRI is expected to increase real income by 1.2-3.4% in BRI economies and 0.7%-2.9% globally, with investments that could lift more than 7.6 million people out of extreme poverty. In particular, BRI transport projects have significantly reduced trade costs, which are expected to increase trade in BRI economies by 2.8% to 9.7%, increase in world trade by 1.7% to 6.2%, and total foreign direct investment in BRI economies by 4.97% (World Bank, 2019). The BRI has strengthened cooperation in trade, investment, infrastructure construction, institutional and cultural exchange among Asia, Europe and Africa, and formed a new economic network (Anwar et al., 2021). It aims to build a community of shared future for mankind, and it is an important platform for international multilateral cooperation and regional integration cooperation. Ultimately, it will make the world economy and society more open, inclusive, balanced and benefit-sharing.[ More details about the BRI could check the website of Belt and Road Portal, https://eng.yidaiyilu.gov.cn/index.htm. ] However, there are still significant political, cultural, social, economic and institutional differences among the BRI economies, especially in terms of economic growth and their policies.[ Such differences are not only between China and BRI economies, but also within BRI economies.  Thus, these transnational differences are a collection of differences between many parties.  ]

Comment 3: Check several typos. Change “BRI economics” to “BRI economies”.

Response 3: We appreciated this important reminder, and Thanks for spotting this typo. Following your suggestions, we changed “BRI economics” to “BRI economies” in this paper, and carefully checked the similar misspellings. 

Comment 4: Implication should be rewritten.

Response 4: Thanks for your excellent comments that could improve the quality of our manuscript. Following your suggestions and detailed comments in PDF manuscript, we had rewritten the implication. Based on your comments, we make the following amendments: 

First, we changed the title of the last part of the paper (page 22) from “Conclusion” to “Conclusion and Discussion”. Second, we summarized the conclusions of this paper and discussed the structural characteristics of BCSN before and after the inception of the BRI. Third, this paper only revealed the structural characteristics of BCSN of BRI economies and did not make regression test on the driving factors of the BCSN, so this paper could not make corresponding policy suggestions. 

So, we deleted the policy implications in the introduction part and the policy recommendations in the last paragraph (page 26), and added the discussion of future research directions. The related revision is shown on pages 22-24 and below: 

With the support of instantaneous quasi-correlation and complex network analysis method, this article empirically analyzed the Topological characteristics of BCSN of BRI economies from 2000 to 2019. The main research conclusions are as follows:  

First, in the sample period, the business cycle synchronization linkage of BRI economies is still relative weak, the network density and network efficiency has decreased after the inception of the BRI.

Second, the BCSN of BRI economies always show a large clustering coefficient and a short average path length, which presents a typical structural characteristic of “small world”. Meanwhile, the clustering degree of output synchronization linkage of BRI economies is weakened after the inception of the BRI.

Third, on the whole, the BCSN of BRI economies does not have the characteristic of gradual evolution. But since the inception of the BRI，the evolution of the BCSN of BRI economies shows a self-stability characteristic.

Fourth, the BCSN structure is composed of four subgroups. The synchronization linkage level inside subgroups is obviously higher than the one outside subgroups. After the inception of the BRI, the BCSN structure of BRI economies mainly consist of two-way spillover subgroup and broker subgroup. 

Fifth, the individual characteristics of the BRI economies are obviously different, the relative influence of a country in the network does not fully show its hub role. After the inception of the BRI, the function of China and other important nodes, such as Southeast Asia, Central Asia and CEE, has been enhanced. From the perspective of different regions, East Asia plays a relatively big role in the network, CEE and Central Asia take the most prominent mediating effect. 

Sixth, China's EC and coreness level have increased and ranked top during the sample period, but its CC level is still low. After the inception of the BRI, China's relative influence in the entire network has increased significantly, but does not show much mediating effect.

Combined with instantaneous quasi-correlation and complex network analysis method, this paper reveals the structural characteristics of the BCSN of BRI economies from the overall and individual aspects. It should be noted that there are still some deficiencies in this paper, which need to be improved by follow-up research. Specifically, future research can be improved from the following three aspects. First of all, besides the instantaneous quasi-correlation method, other methods can be used to measure the business cycle synchronization, such as Markov switching model, dynamic correlation coefficient method, GARCH Model, concordance index and difference method, to ensure the robustness of empirical results. In addition, combined with cutting-edge complex network analysis methods, it is considered to make a weighted business cycle synchronization matrix, and further investigate the characteristics of the BCSN of BRI economies such as robustness and vulnerability, so as to enrich the existing research. Finally, Multiple Regression Quadratic Assignment Procedure (MRQAP) could be used to empirically study the driving factors of the structure evolution of the BCSN of BRI economies, which may provide some policy suggestions for strengthening the business cycle synchronization linkage of BRI economies.

[References]

Anwar M. A., S. Nasreen, and A. K. Tiwari. Forestation, Renewable Energy and Environmental Quality: Empirical Evidence From Belt and Road Initiative Economies[J]. Journal of Environmental Management, 2021, 291(8):112684-112683.

Klinger J. Environment, Development, and Security Politics in the Production of Belt and Road Spaces[J]. Territory Politics Governance, 2019,8(5):657-675.

World Bank. Belt and Road Economics: Opportunities and Risks of Transport Corridors[R]. Washington, DC: World Bank. 2019.

Responses to Reviewer #2: 

Dear reviewer:

Thank you for your comments on our manuscript. Those comments are valuable and very helpful for revising and improving our paper. We have studied the comments carefully and have made corrections which we hope to meet with approval finally.

Below we address each issue the reviewers raised and describe corresponding changes in the revised manuscript. For ease of reference, changes were made by using blue text in the revised manuscript. Reviewers’ comments are repeated in full using italics, while our responses are typed in standard.

1.General Comment: The paper is interesting, its structure is correct, and the topic is clearly a hot one. Results are fine unless not surprising. However, from my point of view, the following main issues make its publication not recommended at its current state.

Response: Thank you for your valuable time and efforts in this review process, and Thanks for your excellent comments that could improve the quality of our manuscript. We significantly improved the quality of this manuscript based on your constructive comments. Here we submitted a new version, which has been further revised according to your suggestions. In summary, major modifications include: 

First, in the first introduction part, we introduced BRI in a more concise way, restated the research questions in this paper, and expounded the research content more accurately.  

Second, in the second part of related literature, we reorganized relevant literature, made some improvements in the measurement of business cycle synchronization, the research on the structural characteristic of the BCSN, and summarized the shortcomings of existing research and the contribution of this paper.  

Third, in the third part of research methods and data description, we mainly improved the description of complex network analysis methods, especially the construction process of BCSN, and added the calculation formula of structural characteristic indicators.  

Fourth, in the fourth part of the overall characteristics analysis, we deleted the redundant results of weighted clustering coefficient (WCC), added measuring results and the analysis before and after the inception of the BRI, redrew the diagram of network subgroup structure with VOSviewer software, and elaborated overall structural characteristics of the BCSN from the network structure evolution and network subgroup structure. 

Fifth, in the part of individual characteristics analysis, we deleted the redundant node strength (NS) results, improved the interpretation of the measurement results before and after the inception of the BRI, and illustrated the individual structural characteristics of the BCSN from node centrality and coreness degree.  

Sixth, in the conclusion and discussion part, we refined the research conclusions, deleted the original policy recommendations, and added the discussion on future research prospects.

Seventh, we tried our best to correct punctuation marks, misspellings, grammar, references and other details in the paper, and deleted the “speculative” statements, in order to improve the quality of this paper and meet relevant requirements.

2.Major comments

Comment 1: Introduction is extremely long and not very informative. At the end of it, we still do not know what the “BRI” is (who is financing the initiative, what for, what the achievements are, what kind of projects have been implemented, and so on)

Response 1: Thanks for your constructive comments that could improve the quality of our manuscript. Following your suggestions, we have simplified and adjusted the content of the introduction. To be specific, first of all, we have rewritten the introduction of BRI to highlight the origin, coverage, achievements, goals and challenges of BRI. Secondly, based on the current challenges BRI is facing, this paper introduces relevant research on business cycle synchronization, and reveals its key issues. Then, in order to reveal the complex interdependence among economies, we introduce the complex network analysis method and its application in analyzing the business cycle synchronization linkage. Finally, taking BRI economies as the research object, we elaborate the problems to be solved in this paper and introduce specific research methods. 

For more detailed modifications, please refer to the introduction part (page 1-4). 

More detialed introduction about BRI could refer to the website of Belt and Road Portal (https://eng.yidaiyilu.gov.cn/index.htm). In addition, in order to facilitate reviewer's understanding, we make a separate summary here.

About the origin of BRI, based on the history of China's ancient Silk Road, Chinese President Xi Jinping proposed the Silk Road Economic Belt and the 21st Century Maritime Silk Road during his visit to Kazakhstan and Indonesia in 2013, thus forming the whole Belt and Road Initiative. As the initiator and leader of BRI, China hoped to strengthen economic cooperation among BRI economies through this initiative and jointly realize sustainable economic and social development. 

About the coverage of BRI, the Belt and Road is composed of more than 50 different economies, covering 80% of the global population with an economic scale of over 21 trillion US dollars (Klinger, 2019). BRI is an effort to create jointly-built trade routes that emulate the ancient Silk Road and promote regional cooperation in Asia, Europe, and Africa.

About the development achievements of BRI, it has strengthened cooperation in trade, investment, infrastructure construction, institutional and cultural exchange among Asia, Europe and Africa, and formed a new economic network (Anwar et al., 2021). Since the inception of the BRI in 2013,a lot of achievements in infrastructure development, international trade, cross-border investment and sustainable growth have been made. One World Bank report[ For a more details of the achievements of the BRI, please refer to the World Bank report Belt and Road Economics:  Opportunities and Risks of Transport Communities,  https://www.worldbank.org/en/topic/regional-  integration/publication/belt-and-road-economics-opportunities-and-risks-of-transport-corridors.  ] noted that BRI is expected to increase real income by 1.2-3.4% in BRI economies and 0.7%-2.9% globally, with investments that could lift more than 7.6 million people out of extreme poverty. In particular, BRI transport projects have significantly reduced trade costs, which are expected to increase trade in BRI economies by 2.8% to 9.7%, increase in world trade by 1.7% to 6.2%, and total foreign direct investment in BRI economies by 4.97%(World Bank, 2019). Based on policy coordination, facilities connectivity, unimpeded trade, financial integration and people-to-people bond, some great achievements have been made since the inception of the BRI in 2013. For policy coordination, at presen, China has signed more than 200 agreements on BRI cooperation with 149 countries and 32 international organizations, and successfully hosted two BRI Forums for international cooperation. For facilities connectivity, infrastructure construction cooperation between China and BRI economies in railways, ports, aviation, energy and communications has strengthened the infrastructure quality of BRI member states. For unimpeded trade, trade and investment among BRI economies have grown considerably, and BRI members have actively participated in the China International Import Expo. For financial integration, the Silk Road Fund and the Asian Infrastructure Investment Bank were set to provide reliable financial support for BRI construction projects. For people-to-people bond, China and BRI economies have deepened cooperation in cultural exchanges, scientific and technological innovation, and conducted a series of humanitarian assistance in medical care, poverty alleviation and food supply.

About development goals of BRI, it aims to build a community of shared future, and it is an important platform for international multilateral cooperation and regional integration cooperation. Ultimately, it will make the world economy and society more open, inclusive, balanced and benefit-sharing.[ More details about the BRI could check the website of Belt and Road Portal,https://eng.yidaiyilu.gov.cn/index.htm.]

About the challenge BRI is facing, there are still significant political, cultural, social, economic and institutional differences among the BRI economies, especially in terms of economic growth and their policies[ Such differences are not only between China and BRI economies, but also within BRI economies.  Thus, these transnational differences are a collection of differences between many parties.  ]. Just because of the difference in economic base and growth, it poses certain challenges to promote BRI development.

Comment 2: The paper, from my point of view, is very poorly written and difficult to understand. There are plenty of examples, I will show just a couple of them:

• Page 8, “With the continuous advancement of economic globalization and international division of labor, the correlation trend of output periodicity and economic scale among countries presents a certain nonlinear and multi-threaded complex relationship, and then forms an interactive BCSN in space”

• Page 9, “are regarded as nodes in this paper, and the quasi-correlation coefficients between the actual economic scales of economics are regarded as the connecting edges between nodes”

Response 2: Thanks for your excellent comments, and we appreciated this important reminder. Following your suggestions, we tried our best to improve the English expression and grammar in this paper in order to meet the relevant requirements as far as possible.  

As the example sentence on page 8 you mentioned, after careful consideration, we have deleted it directly. According to Antonakakis et al.(2015) and Papadimitriou et al.(2016), we constructed BCSN directly using complex network analysis methods without additional explanation.   

As you mentioned on page 9, after careful analysis, we have changed the sentence as: “According to complex network theory and graph theory, BRI economies are regarded as nodes, and the business cycle synchronization (i.e. the value of instantaneous quasi-correlation) between economies are regarded as edges between nodes.”

Comment 3: Figures throughout the paper are plotted in an extremely low quality and they are no informative.

Response 3: Thanks for your excellent point. Following your suggestions, we had deleted the original figures, and redrew it by VOSviewer software. The new figures are shown on pages 16 and below:

Fig. 1. Network Subgroups during 2000-2013.

Source: Drawn by the authors from VOSviewer software.

Fig. 2. Network Subgroups during 2014-2019.

Source: Drawn by the authors from VOSviewer software.

Comment 4: The paper shows an important number of “speculative” statements, especially when it comes to policy advice. 

For instance, page 13, the authors claim: “It is worth noting that in 2019, network density and efficiency saw a significant decline, which was mainly caused by the unilateral trade protectionism and continuous trade frictions among western countries led by the US in recent years, which worsened the previous positive global economic and trade environment”. However, the authors do not analyse the variables affecting output synchronicity and consequently they cannot state such connection. 

In page 4 the authors claim that their paper will provide policy inspiration to enhance economic synchronicity among BRI countries, however this is not possible with the analysis the authors provide.

Response 4: Thanks for your constructive comments. It should be noted that the purpose of this study is to reveal the structural characteristics of the BCSN, and the driving factors of BCSN are not further investigated through regression test. Therefore, this paper is indeed unable to explain the results of BCSN structural characteristics, let alone make corresponding policy suggestions based on the existing results. In view of this, as you mentioned, we have deleted the “speculative” statements on page 4 and page 13 mentioned by experts, and re-examined and revised the “speculative” statements in the part of introduction, structural characteristics analysis and policy suggestions in the paper.

In addition, it is worth noting that we have changed the title of the last part of the paper (page 22) from “Conclusion” to “Conclusion and Discussion”. And then, we summarized the conclusions of this paper and discussed the structural characteristics of BCSN before and after the inception of the BRI. In the end, we deleted the policy recommendations in the last paragraph (page 23-24), and added the discussion of future research directions.    

The related revision is shown on pages 22-24 and below: 

With the support of instantaneous quasi-correlation and complex network analysis method, this article empirically analyzed the Topological characteristics of BCSN of BRI economies from 2000 to 2019. The main research conclusions are as follows:  

First, in the sample period, the business cycle synchronization linkage of BRI economies is still relative weak, the network density and network efficiency has decreased after the inception of the BRI.

Second, the BCSN of BRI economies always show a large clustering coefficient and a short average path length, which presents a typical structural characteristic of “small world”. Meanwhile, the clustering degree of output synchronization linkage of BRI economies is weakened after the inception of the BRI.

Third, on the whole, the BCSN of BRI economies does not have the characteristic of gradual evolution. But since the inception of the BRI，the evolution of the BCSN of BRI economies shows a self-stability characteristic.

Fourth, the BCSN structure is composed of four subgroups. The synchronization linkage level inside subgroups is obviously higher than the one outside subgroups. After the inception of the BRI, the BCSN structure of BRI economies mainly consist of two-way spillover subgroup and broker subgroup. 

Fifth, the individual characteristics of the BRI economies are obviously different, the relative influence of a country in the network does not fully show its hub role. After the inception of the BRI, the function of China and other important nodes, such as Southeast Asia, Central Asia and CEE, has been enhanced. From the perspective of different regions, East Asia plays a relatively big role in the network, CEE and Central Asia take the most prominent mediating effect. 

Sixth, China's EC and coreness level have increased and ranked top during the sample period, but its CC level is still low. After the inception of the BRI, China's relative influence in the entire network has increased significantly,but does not show much mediating effect.

Combined with instantaneous quasi-correlation and complex network analysis method, this paper reveals the structural characteristics of the BCSN of BRI economies from the overall and individual aspects. It should be noted that there are still some deficiencies in this paper, which need to be improved by follow-up research. Specifically, future research can be improved from the following three aspects. First of all, besides the instantaneous quasi-correlation method, other methods can be used to measure the business cycle synchronization, such as Markov switching model, dynamic correlation coefficient method, GARCH Model, concordance index and difference method, to ensure the robustness of empirical results. In addition, combined with cutting-edge complex network analysis methods, it is considered to make a weighted business cycle synchronization matrix, and further investigate the characteristics of the BCSN of BRI economies such as robustness and vulnerability, so as to enrich the existing research. Finally, Multiple Regression Quadratic Assignment Procedure (MRQAP) could be used to empirically study the driving factors of the structure evolution of the BCSN of BRI economies, which may provide some policy suggestions for strengthening the business cycle synchronization linkage of BRI economies.

Comment 5: References regarding network analysis applied to synchronization is quite poor. The authors just cite 3 or 4 papers dealing with this issue but many more can be found and should be referenced in the paper. In the same line, the authors have not deeply revised the economic literature on business cycle synchronization.

Response 5: Thanks for your excellent comments that could improve the quality of our manuscript. Following your suggestions, after carefully re-reading and sorting out relevant literature, literature closely related to this paper mainly fall into the following two categories: One is the measurement research on the international business cycle synchronization (literature from economics view), the other is the related research on the BCSN (literature about network analysis ). Therefore, we readjusted and improved the related literature part, adding not only the economic literature on the measurement of business cycle synchronization, but also the literature on the structural characteristics of the BCSN. Specific modifications can be found in the related literature part on pages 5-8 in the paper.

In order to facilitate the review by reviewers, we presented the revised related literature part as: 

The literature closely related to this paper mainly includes the following two categories: one is the measurement and research on the international business cycle synchronization; the other is on the BCSN.

The measurement and research on international business cycle synchronization mainly includes static method and dynamic method.  For the static methods, existing studies mainly adopt simple correlation coefficient method (Giovanni and Levchenko, 2010; Papadimitriou et al., 2014) and dynamic factor model (Del Negro and Otrok, 2008; Kose et al., 2012). It should be noted that although the static method can intuitively judge the level of the synchronization, it cannot reveal the dynamic characteristics of the business cycle synchronization. Therefore, the follow-up measurement research has gradually shifted from the traditional static methods to the dynamic methods. For dynamic methods, existing studies mainly focus on Markov Regime Switching Model (Hamilton and Owyang, 2012; Leiva,Leon, 2014; Ductor and Leva-Leon, 2016), GARCH model (Savva et al., 2010; Antonakakis,2012), Concordance Index (Harding and Pagan, 2006; Cerqueira and Martins, 2009; Cerqueira, 2013), Difference Method (Kalemli-Ozcan et al., 2013;  Pyun and An, 2016) and instantaneous quasi-correlation method (Abiad et al., 2013;  Duval et al., 2016). In addition, some scholars used the Method of Hodrick-Prescott (HP) filter to de-trend the original output data series and further calculate the business cycle synchronization (Ng, 2010; Huang and Zhu, 2015). However, Hamilton (2018) believes that HP filtering method introduces unreal dynamic linkages that are not based on original data, and its results are affected by the size of smoothing parameter, so it cannot truly reflect the level of business cycle synchronization. Compared with other dynamic methods, the instantaneous quasi-correlation method is more efficient and could carry out dynamic calculation and analysis, which avoids problems caused by artificial parameter setting and distortion of original data. Therefore, it has been widely used in practical calculation and analysis (Abiad et al., 2013;  Yao and Tang, 2020). 

As for the research on the BCSN, the existing research is mainly conducted from the perspective of network analysis. Diebold and Yilmaz(2013) investigated the linkages between actual outputs of G7 countries from 1962 to 2010 by network analysis, and found that the indicator of connectedness(density) could measure the pairwise output fluctuation linkages between different countries. At the same time, global connectedness will change as the business cycle changes. Gomez et al.(2013) adopted correlation coefficient matrix and network analysis method to systematically investigate the co-movement of business cycles synchronization in various countries since 1950, and believed that the dynamic changes of interdependence among countries was mainly driven by the co-movement of regional economic growth rather than co-movement of world economy. Caraiani (2013) found that compared with the Granger causality method, using correlation coefficients to construct a directed business cycle synchronization network can reflect the relative influence of countries in the world economy more reasonably , and further empirical findings showed that the United States finally occupies the core position of the business cycle synchronization network of G7 and OECD. Papadimitriou et al. (2014) used Pearson correlation coefficient and minimum dominating set to make an empirical study on the BCSN structure of 22 EU sample member states, and found that after the adoption of the common currency euro, the output of member states had a higher correlation, and the BCSN density of EU was increasing. Xi et al. (2014) constructed the BCSN of G7 based on the pairwise Maximum Entropy Model, and found that the network presented a clustering hierarchy and nearly accounted for almost half of the entire structure of the interactions within the G7 system. Antonakakis et al. (2015) used sign concordance index and threshold-minimum dominating set method to investigate topological characteristcs of BCSN among 27 countries during 1875-2013 and find that there are obvious differences in node degree of different countries in different periods. Gomez et al. (2017) used correlation coefficient and minimum spanning tree technique to construct the BCSN of EU, analyzed the business synchronization linkages and accessibility among member states, and found that there was no obvious core-periphery structure inside the network. Ductor and Leva-Leon (2016) adopted the social network analysis method and the indicator of betweenness centrality to evaluate the relative influence of various countries for global BCSN. It is found that a country's is more influential in the network tends to increase when the economy is in recession, but becomes less influential when the economy is expanding. With Pearson correlation coefficient, rolling window, threshold-minimum dominating set and other methods, Papadimitriou et al. (2016) selected some indicators such as the total number of peripheries, network density, the number of dominant and isolated nodes and node degree to empirically investigate the topological characteristics of European BCSN during 1986-2011. Matesanz and Ortega(2016) used similarity index and Minimum Spanning Tree technique (MST) to construct the European BCSN from 1950 to 2013. Based on the dynamic network analysis, the correlations and connectivity of the network increased significantly in 2009. In addition, differences in window size, filtering method and similarity measure also affect the characteristics of the BCSN. With the help of correlation coefficient index used by Cerqueira (2013), Belke et al. (2017) investigated the the business cycle synchronization of the European Monetary Union, focusing on the core-periphery mode of the business cycle within the Union after the economic crisis. Leiva-Leon (2017) established American inter-state BCSN by Markov Regime Switching framework and investigated the its evolution model with indicators of MDS (multidimensional scaling) and closeness centrality. It is found that the network has an obvious core-periphery structure. Sebestyén and Iloskics (2020) employed the pairwise Granger causality between national outputs to construct the global shock contagion network, and found that it has a relative long path length and stronger transmission, and the degree distribution tends to be asymmetric. 

For the research on the BSCN of BRI economies, Huang and Yao (2018) used CM (Cerqueira & Martins) synchronization index to measure the business cycle synchronization between China and BRI economies and find that it shows a certain “decoupling” trend, and there are obvious differences in different stages and different development aspects. Cui et al. (2020) used dynamic correlation coefficient to calculate and found that China and Southeast and Central Asian countries, as well as Mongolia, Nepal, Pakistan, Sri Lanka and other countries have a high business cycle synchronization level. Du et al. (2020) found that the BRI strengthened the business cycle synchronization linkages between China and ASEAN countries, while the network density and clustering coefficient also increased to some extent.

Comment 6: I do not understand Table 1. In it the Networks density measure is provided. This measure is shown for different years. However, if I am correct, the edges of the network represent correlation degree while nodes represent countries. Therefore, what is the meaning of this measure for year 2000, which is the first year of analysis? Or the same measure for 2019? This is not clear to me.

Response 6: Thanks for your point. As you proposed, we used the quasi-correlation degree and nodes to represent the periphery of the network and the BRI economies respectively. In the complex network analysis part of this paper, we gave the meaning of the structural characteristics indicators, and then calculate the results of the annual network structural characteristics. At this point, we intended to show the results of overall structural characteristics on the BCSN of BRI economies in some years in Table 1, including density (DS), network efficiency (NE), clustering coefficient (CC), average distance (AD) and other indicators. Before explaining this problem, we have added the formula and definition of each indicator for your understanding.

In this paper, density (DS) describes the degree of business cycle synchronization between the BRI economies. For better understanding, we added its formula as ，where represents the actual number of effective linkages in the BCSN in year . represents the nodes in the network (). It should be noted that the DS is calculated by annual unweighted and undirected BCSN based on a symmetric Instantaneous Quasi-correlation matrix. At this time, for the BCSN in year , the actual number of effective linkages in the network equals to the actual effective business cycle synchronization linkages, while in theory, the maximum linkage number equals to =53*52=2756. Furthermore, when the instantaneous quasi-correlation between two economies is greater than the average among all economies, it is indicated that the business cycle synchronization linkages between the two economies is actually effective, and it will be recorded as an actual effective linkage number. Since the output growth and growth rate of BRI economies vary significantly every year, the size of the instantaneous quasi-correlation among economies will also change, leading to changes in the actual number of effective linkages in the network. Meanwhile, the constant number of nodes in the network is 53, so theoretically the maximum linkage number does not change.     

To sum up, no matter in 2000, 2019 or any other year, DS is calculated in the same way with the same meaning, that is, the ratio of the actual effective number of linkages in the network to the theoretical maximum number of linkages in year t .

[References]

Abiad A., D. Furceri S. Kalemli-Ozcan, and A. Pescatori. Dancing Together? Spillovers, Common Shocks, and the Role of Financial and Trade Linkages[M]. Washington: International Monetary Fund,World Economic Outlook 2013.

Antonakakis N. Business Cycle Synchronization During US Recessions Since the Beginning of the 1870s[J]. Economics Letters, 2012,117(2):467-472.

Anwar M. A., S. Nasreen, and A. K. Tiwari. Forestation, Renewable Energy and Environmental Quality: Empirical Evidence From Belt and Road Initiative Economies[J]. Journal of Environmental Management, 2021, 291(8):112684-112683.

Belke A., Domnick C., Gros D. Business Cycle Synchronization in the EMU: Core vs. Periphery[J]. Open Economies Review, 2017, 28(5):1-30.

Caraiani P. Using Complex Networks to Characterize International Business Cycles[J]. PLOS ONE, 2013,8(3), e58109:1-13.

Cui, Q. Y., Y. Zhang, and S. Wang. International Macroeconomic Policy Coordination Under the BRI: Mechanism Basis and China’s Role[J]. Economists, 2020(8):49-58.

Del Negro, M., and C. Otrok. Dynamic Factor Models with Time-Varying Parameters: Measuring Changes in International Business Cycles[C]. Federal Reserve Bank of New York Staff Reports, No. 326. 2008.

Diebold F. X., Yilmaz K. Measuring the Dynamics of Global Business Cycle Connectedness [C]. PIER Working Paper No.13-070, 2013.

Du, J. Z., C. Z., Wang, and J. L. Wang. An Analysis of Business Cycle Synchronization Between China and ASEAN Countries in the Context of the Belt and Road Initiative[J]. Journal of Finance and Economics Theory, 2020(4):23-31.

Ductor L, Leiva-Leon D. Dynamics of Global Business Cycles Interdependence[J]. Journal of International Economics, 2016, 102:110-127.

Duval R, Li N, Saraf R, Seneviratnea D. Value-added trade and business cycle synchronization[J]. Journal of International Economics, 2016,99:251-262．

Giovanni D, J., Levchenko G. Putting the Parts Together: Trade, Vertical Linkages and Business Cycle Comovement[J]. American Economic Journal Macroeconomics, 2010, 2(2):95-124.

Gomez D. M., Torgler B., Ortega G. J. Measuring Global Economic Interdependence: A Hierarchical Network Approach[J]. The World Economy, 2013,36(12):1632-1648.

Gomez D. M, Ferrari H. J, Torgler B, Guillermo J. O. Synchronization and Diversity in Business Cycles: A Network Analysis of the European Union[J]. Applied Economics, 2017,49(10):1-15.

Hamilton J. D. Why You Should Never Use the Hodrick-Prescott Filter[J]. Review of Economics and Statistics, 2018,100(5):831-843.

Hamilton J. D. M. Owyang. The Propagation of Regional Recessions[J]. Review of Economics and Statistics, 2012, 94(4):935-947.

Harding D., Pagan A. Synchronization of Cycles[J]. Journal of Econometrics, 2006,132(1):59-79.

Huang Z. L., Yao T. T. Business Cycle Synchronization and its Transmission Mechanism between China and the Countries along the BRI[J]. Statistical Research, 2018,35(9):40-53.

Huang Z. L., Zhu B. H. Real Business Cycle and Taxation Policy Effects in China[J]. Economic Research Journal, 2015,50(3):4-17+114.

Kalemli-Ozcan S., Papaioannou E., Perri F. Global Banks and Crisis Transmission[J]. Journal of International Economics, 2013, 89:495-510.

Klinger J. Environment, Development, and Security Politics in the Production of Belt and Road Spaces[J]. Territory Politics Governance, 2019,8(5):657-675.

Kose M. A., Otrok C., and E. Prasad. Global Business Cycles: Convergence or Decoupling?[J]. International Economic Review, 2012,53(2):511-538.

Leiva-Leon D. Measuring Business Cycles Intra-Synchronization in US: A Regime-switching Interdependence Framework[J]. Oxford Bulletin of Economics & Statistics, 2017,79(4):513-545.

Matesanz D., Ortega G. J. On Business Cycles Synchronization in Europe: A Note on Network Analysis[J]. Physica A: Statistical Mechanics and its Applications. 2016; 462:287-296.

Ng E. Y. Production Fragmentation and Business-cycle Co-movement[J]. Journal of International Economics, 2010, 82(1):1-14.

Papadimitriou T, Gogas P, Sarantitis G A. European Business Cycle Synchronization: A Complex Network Perspective, Network Models in Economics and Finance[M]. 2014, in Chapter 13:265-275.

Papadimitriou T., Gogas P., Sarantitis G. A. Convergence of European Business Cycles: A Complex Networks Approach[J]. Computational Economics, 2016,47:97-119.

Pyun J. H., An B. J. Capital and Credit Market Integration and Real Economic Contagion During the Global Financial Crisis[J]. Journal of International Money & Finance, 2016, 67:172-193.

Savva C. S., Neanidis K. C., Osborn D. R. Business Cycle Synchronization of the Euro Area with the New and Negotiating Member Countries[J]. International Journal of Finance & Economics, 2010, 15(3):288-306.

Sebestyén T., and Z. Iloskics. Do Economic Shocks Spread Randomly?: A Topological Study of the Global Contagion Network[J]. PLOS ONE, 2020,15:e0238626.

Xi N, Muneepeerakul R, Azaele S, Wang Y G. Maximum Entropy Model for Business Cycle Synchronization[J]. Physica A: Statistical Mechanics and its Applications, 2014,413:189-194.

Yao W., and A. D. Tang. Financial Integration and Business Cycle Synchronization[J]. China Journal of Economics 2020,7(2):61-85.

---

## [Decision Letter · Decision Letter 1]

3 Jun 2022

PONE-D-21-31756R1Topological Characteristics of International Business Cycle Synchronization: A Network Analysis of the BRI EconomiesPLOS ONE

Dear Dr. Sichao,

Thank you for submitting your manuscript to PLOS ONE. We have received two reviewer reports. Both reviewers now recommend publication of the manuscript. However, Reviewer #1 provided a number of minor comments in the attached file. Thus, before formally accepting your manuscript for publication, we encourage you to consider these comments carefully, revise the manuscript accordingly, and resubmit.

We look forward to receiving your revised manuscript.

Kind regards,

Takashi Nishikawa, Ph.D.

Academic Editor

PLOS ONE

Journal Requirements:

Reviewers' comments:

Reviewer's Responses to Questions

**Comments to the Author**

1. If the authors have adequately addressed your comments raised in a previous round of review and you feel that this manuscript is now acceptable for publication, you may indicate that here to bypass the “Comments to the Author” section, enter your conflict of interest statement in the “Confidential to Editor” section, and submit your "Accept" recommendation.

Reviewer #1: (No Response)

Reviewer #2: All comments have been addressed

2. Is the manuscript technically sound, and do the data support the conclusions?

Reviewer #1: (No Response)

Reviewer #2: (No Response)

3. Has the statistical analysis been performed appropriately and rigorously? 

Reviewer #1: (No Response)

Reviewer #2: (No Response)

4. Have the authors made all data underlying the findings in their manuscript fully available?

Reviewer #1: (No Response)

Reviewer #2: (No Response)

5. Is the manuscript presented in an intelligible fashion and written in standard English?

Reviewer #1: (No Response)

Reviewer #2: (No Response)

6. Review Comments to the Author

Reviewer #1: Dear Authors

Please note that the changes in the revised version must be highlighted in red color so that reviwers can easily recongize them without time consuming work. It is professional way. Unfortunately, I can't see them in the revised version. I would like to decline review not because of the quality of your paper but because of formality issue.

Reviewer #2: (No Response)

7. PLOS authors have the option to publish the peer review history of their article (what does this mean?). If published, this will include your full peer review and any attached files.

Reviewer #1: No

Reviewer #2: No

---

## [Author Response · Author response to Decision Letter 1]

5 Jun 2022

Responses to Reviewer #1: 

Dear reviewer:

Thank you for your comments on our manuscript. Those comments are valuable and very helpful for revising and improving our paper. We have studied the comments carefully and have made corrections which we hope to meet with approval finally. Below we response each issue that the reviewers raised, and describe corresponding changes by using red text in the revised manuscript.

Following you detailed comments in PDF manuscript, we improved the quality of this manuscript based on your constructive comments. Here we submitted a new version, which has been further revised according to your suggestions. In summary, major modifications include: 

According to your advice, we have revised the abstract on page 1, footnote of introduction on page 3 and the format of references on page 25-29. In addition, we also checked the content of other parts of this paper and corrected the errors found.

In the end, it should also be noted that in the last paragraph of page 10 and the formula (2) on page 11, due to the mistake in the last revised manuscript, we misused the row mean of the matrix as the threshold value for data processing. In fact, we should use the mean of the matrix as the threshold value instead, and the undirected and weighted BCSN could be unweighted to obtain the adjacency matrix A composed of 0 and 1. In view of this, we have modified the relevant explanations in the last paragraph of page 10 and the formula (2) on page 11. The above modifications do not affect the reliability and validity of the empirical results of this paper.

---

## [Editor Report · Decision Letter 2]

9 Jun 2022

Topological Characteristics of International Business Cycle Synchronization: A Network Analysis of the BRI Economies

PONE-D-21-31756R2

Dear Dr. Sichao,

We’re pleased to inform you that your manuscript has been judged scientifically suitable for publication and will be formally accepted for publication once it meets all outstanding technical requirements.

Kind regards,

Takashi Nishikawa, Ph.D.

Academic Editor

PLOS ONE

Additional Editor Comments (optional):

Thank you for answering the final question regarding the threshold used to process the quasi-correlation matrices.
---

## [Editor Report · Acceptance letter]

20 Jun 2022

PONE-D-21-31756R2 

Topological Characteristics of International Business Cycle Synchronization: A Network Analysis of the BRI Economies 

Dear Dr. Sichao:

I'm pleased to inform you that your manuscript has been deemed suitable for publication in PLOS ONE. Congratulations! Your manuscript is now with our production department. 

Kind regards, 

on behalf of

Dr. Takashi Nishikawa 

Academic Editor

PLOS ONE